# Exciton Transfer Dynamics and Annihilation in Rubidium–Cesium-Alloyed, Quasi-Two-Dimensional Perovskite

**Lamiaa Abdelrazik** [1], **Vidmantas Jašinskas** [1], **Žydrūnas Podlipskas** [2], **Ramūnas Aleksiejūnas** [2], **Gintautas Tamulaitis** [2], **Vidmantas Gulbinas** [1] and **Aurimas Vyšniauskas** [1,*]

[1]  Center for Physical Sciences and Technology, Saulėtekio Ave. 3, 10257 Vilnius, Lithuania
[2]  Institute of Photonics and Nanotechnology, Vilnius University, Saulėtekio Ave. 3, 10257 Vilnius, Lithuania
[*]  Correspondence: aurimas.vysniauskas@ftmc.lt

**Abstract:** Light-emitting diodes (LEDs) based on perovskite materials are a new group of devices that are currently undergoing rapid development. A significant fraction of these devices is based on quasi-2D perovskites fabricated with large organic cations. In this work, we describe the ultrafast scale dynamics in a quasi-2D $PEA_2(Rb_{0.6}Cs_{0.4})_2Pb_3Br_{10}$ perovskite material with an excess of RbBr, which was previously used to fabricate blue-emitting perovskite LEDs. The results obtained using transient absorption spectroscopy are consistent with the assumption that the carrier dynamics in this material are dominated by excitons, most of which decay by exciton–exciton annihilation when high-intensity excitation is used. Furthermore, a slow energy transfer between different quasi-2D domains taking place within 50 ps was observed. The content of the RbBr did not show any strong influence on the observed dynamics. Our results show that the exciton–exciton annihilation proceeds much faster in thin ($n = 2$) quasi-2D domains than in thick ($n \geq 4$) domains. This finding implies that perovskites with high-$n$, quasi-2D domains are preferable for efficient perovskite lasers and bright perovskite LEDs.

**Keywords:** exciton dynamics; lead-based perovskites; transient absorption spectroscopy



## 1. Introduction

Perovskite light-emitting diodes (PeLEDs) are promising optoelectronic devices due to their unique advantages, such as superior color purity, high efficiency, narrow bandwidth, broadly tunable optical band gaps, structural diversity, and facile solution-processability [1–7]. Therefore, the scientific community's interest in PeLEDs has increased rapidly in recent years, thanks to their advantages over organic light-emitting diodes (OLEDs) and inorganic quantum-dot LEDs (QD-LEDs) [8–12]. In 2014, Snaith et al. were the first to report the maximum external quantum efficiency (EQE) of 0.1% for an organic–inorganic PeLED [13]. Since then, PeLEDs have been intensively developed to reach EQEs as high as 20.7% [14], 21.3% [15], 45.5% [16], and 13.3% [17] for near-infrared, red, green, and blue PeLEDs, respectively, which are comparable to those of state-of-the-art OLEDs and QD-LEDs [5,14,15,18–24].

Quasi-2D lead perovskites are used as emitting layers for a large number of the currently known PeLEDs [25,26]. These materials have large exciton-binding energies, leading to fast radiative decay and high photoluminescence quantum yields (PLQY). Finally, they are relatively stable in the ambient environment. Quasi-2D perovskite materials can be prepared by modifying the chemical formula of 3D lead halide perovskites ($APbX_3$), where A is a monovalent cation, such as methylammonium ($MA^+$), formamidinium ($FA^+$), or $Cs^+$, and $X^-$ is a halide ion (i.e., $Cl^-$, $Br^-$, $I^-$, or their mixtures), by replacing the small organic cation with a large organic cation (R) to form perovskites with the chemical formula $R_2A_{n-1}B_nX_{3n+1}$. Here, $n$ represents the number of planar layers composed out of $PbX_6$ octahedra [27], and $n = 2$ corresponds to a 2D perovskite structure, whereas a 3D perovskite

corresponds to $n = \infty$. In the intermediate case of $n \geq 2$, the perovskite is referred to as a quasi-2D perovskite (Figure 1). Typical large cations used to form quasi-2D perovskites are phenylethylammonium (PEA) or butylammonium (BA), which have a low dielectric constant and act as barriers between the perovskite layers for free carriers and excitons, creating a quantum well (QW) structure. A typical quasi-2D perovskite film contains multiple quasi-2D domains with different $n$-values and different band gaps [28]. When the exciton in the film is formed, either by photoexcitation or by the recombination of injected carriers, a process called "exciton funneling" takes place from the domains with low $n$-values and large band gaps to the domains with high $n$-values and low band gaps [4].

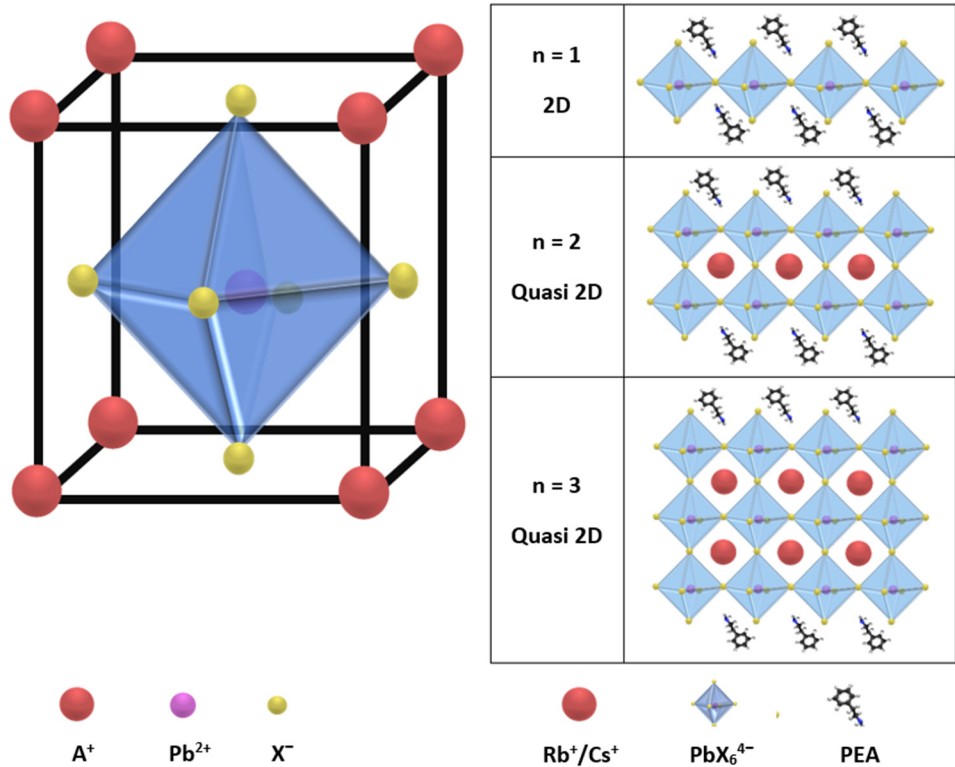

**Figure 1.** Schematic diagram of $APbX_3$ lead halide perovskite crystal structure (**left**); 2D and quasi-2D perovskite with $n$-values from 1 to 3 (**right**). Octahedra denote $PbX_6^{4-}$ fragments, and small circles stand for $Rb^+$ or $Cs^+$ ions, whereas PEA is indicated by its molecular shape.

Recently, several studies have been carried out to investigate the exciton and free-carrier dynamics in quasi-2D perovskites [29–36]. This knowledge is important because the enhanced probability of many-body interactions, such as exciton–exciton annihilation or Auger recombination in low-dimensional materials. In such a case, the nonradiative recombination rate is higher, and the overall performance of LEDs based on quasi-2D perovskites deteriorates as a result. The exciton-binding energies of quasi-2D perovskites reach several hundred MeV [37,38], which is much greater than the thermal energy at room temperature. Consequently, the nonequilibrium-carrier dynamics will be dominated by excitons. Several studies have been carried out with a focus on the annihilation of excitons at different excitation-energy densities in quasi-2D domains of different thicknesses [39,40] or on exciton transport [30]. However, several works show that the excitons, despite their large exciton-binding energies, can rapidly dissociate into free carriers due to the presence of localized states at the edge of a perovskite layer [35,41]. Moreover, the excitons can be transferred to another quasi-2D domain by temporarily splitting into free carriers, which are transferred sequentially, reforming the exciton in the new domain [29]. Finally, it has been calculated that the thickness of the quasi-2D sheet becomes greater than the exciton Bohr radius when $n$ is about 4–6. Thus, the exciton dynamics in such sheets should

be similar to those in a 3D perovskite. In summary, both excitons and free carriers may dominate the dynamics in quasi-2D perovskites depending on the domain thickness, excitation intensity, exciton density, and temperature, complicating the picture and warranting further investigation.

In this work, we investigated the carrier dynamics taking place in excited Rb–Cs-alloyed lead bromide quasi-2D perovskites with and without excess RbBr. The chemical formula is $PEA_2(Rb_{0.6}Cs_{0.4})_2Pb_3Br_{10}$, which corresponds to domains with $<n> = 3$. Previously, such perovskites with a 100% excess of RbBr have been used as emissive layers for blue PeLEDs [26], which were among the most efficient at the time. Using an excess of Cs or Rb halides in the fabrication of perovskites increases their PLQY [26,42,43], making them more suitable for LEDs. However, to the best of our knowledge, the effects of such an excess on the exciton and free-carrier dynamics in perovskites have not been investigated thus far. Furthermore, we also studied the differences in the dynamics observed in quasi-2D domains with different $n$-values. For this purpose, we used transient absorption (TA) spectroscopy at different temperatures. Our results support the assumption that the dynamics in $PEA_2(Rb_{0.6}Cs_{0.4})_2Pb_3Br_{10}$ are dominated by excitons. The main pathway of their relaxation is exciton–exciton annihilation, which proceeds much faster in thin $n = 2$ domains, while the excess of RbBr has no strong influence.

## 2. Materials and Methods

### 2.1. The Preparation of the Perovskite Solution

To achieve the targeted composition of a quasi-2D perovskite, we prepared a perovskite solution by continuous stirring, heating (if necessary), and dissolving overnight 0.33 M phenyl-ethyl ammonium bromide (PEABr) ($\geq$98% purity, Sigma-Aldrich, USA), 0.50 M lead bromide ($PbBr_2$) (99.99% purity, Sigma-Aldrich, USA), and 0.13 M cesium bromide (CsBr) (99.99% purity, Sigma-Aldrich, USA), with different excess concentrations (0.20–0.40 M) of rubidium bromide (RbBr) (99.6% purity, Sigma-Aldrich, USA) in anhydrous dimethyl sulfoxide (DMSO) (99.7% purity, extra dry, Acros Organics, USA). The resulting molecular formula of the perovskite compound was $PEA_2(Rb_{0.6}Cs_{0.4})_2Pb_3Br_{10}$, corresponding to the $<n> = 3$ quasi-2D perovskite phase, with either a 0%, 50%, or 100% excess of RbBr.

### 2.2. Perovskite Film Fabrication

First, the glass substrates were cleaned using a detergent solution, tap water, and distilled water, and then they were dried with a nitrogen drying gun. To convert the surface of the substrates from hydrophobic to hydrophilic, these films were treated with an oxygen plasma for 10 min using a EMSCD050 plasma cleaner (Leica, Germany), and then they were transferred to a nitrogen-filled glovebox for perovskite deposition.

The fabrication was performed using the solution-based process by using the spin-coating technique, which led to the formation of a thin perovskite film on a glass substrate. An antisolvent solution was used during the spin-coating procedure to facilitate the crystallization of the perovskite film. The solution consisted of triphenylphosphine oxide (TPPO) dissolved in chloroform at a 10 mg/mL concentration. Such an antisolvent solution was previously reported to reduce the number of defects in a perovskite layer and to increase the photoluminescence quantum yield (PLQY) of the film [25]. The perovskite solution was coated onto the substrate through a consecutive, two-step, spin-coating process (1000 rpm for 10 s and 8000 rpm for 30 s, respectively). A quantity of 200 µL of antisolvent was dropped 20 s after the start of the second stage. The perovskite films were thermally annealed at 100 °C for 3 min to remove the excess solvent and accelerate crystallization. After this procedure, the uniformly thin perovskite films were ready for characterization.

### 2.3. Absorbance, Photoluminescence (PL), and Transient Absorption (TA) Measurements

The absorption spectra of the perovskite films were recorded at room temperature by using a spectrophotometer V-670 (JASCO, Germany). The photoluminescence (PL) spectra were acquired using an Edinburgh FLS920 (Edinburgh Instruments, UK) luminescence



spectrometer under excitation at a wavelength of 375 nm. In the ultrafast transient absorption (TA) experiments, the thin perovskite films were excited by laser pulses with a 415 nm wavelength and a 230 fs (full width at half-maximum (FWHM)) pump pulse generated by a Pharos femtosecond laser system equipped with a collinear optical parametric amplifier Orpheus (Light Conversion Ltd., Lithuania). The excitation-energy density ranged from 1.43 $\mu$J/cm$^2$ to 28.6 $\mu$J/cm$^2$. A broadband (400–730 nm) probe beam was produced by the fundamental, 1030 nm wavelength Pharos laser beam by the means of white-light-continuum generation in a sapphire crystal. The duration of the probe pulse was similar to that of the pump pulse. The time delay between the pump and probe pulses was varied by the optical delay line based on retroreflector optics mounted on an electromechanical translation stage (Aerotech PRO165LM, Aerotech Ltd., UK). The TA measurements were carried out in a vacuum at room temperature by placing the sample in a cryostat (DE-204PF, Advanced Research Systems Inc., USA). A spectrometer (Andor Shamrock SR-500i-B1-R, Andor Technology Ltd., UK), equipped with a temperature-controlled CCD camera (Andor Newton DU970P-UVB, Andor Technology Ltd., UK) was used to register the TA spectra. The experiment was managed, and the data were recorded and processed using home-written software in the LabView programming environment. The changes in the absorption ($\Delta A$) were measured as a function of both the wavelength and time delay between pump and probe pulses.

### 2.4. Cathodoluminescence Imaging

Cathodoluminescence (CL) imaging was performed at room temperature using a hybrid cathodoluminescence–scanning electron microscope, Attolight (Switzerland) Chronos, operating in continuous-wave mode. The e-beam was accelerated to 4 keV, the sample was scanned in 18-nanometer steps with the dwell time set to 6.4 ms. The collected light was dispersed using a Horiba (Japan) iHR spectrometer with a grating of 150 grooves/mm density and 500-nanometer blaze wavelength. The CL signal—one spectrum per each pixel of the image—was recorded using an Andor Newton (UK) UV–VIS CCD camera. Data postprocessing and analysis were carried out using Digital Surf's AttoMap (Attolight, Switzerland platform).

### 2.5. Data Analysis

Data processing and visualization were done using OriginPro 9.8 (OriginLab, USA).

## 3. Results and Discussion

### 3.1. Steady-State Spectroscopy

The optical absorption and photoluminescence spectra of the Rb–Cs-alloyed perovskite thin film with a 0% (black), 50% (red), and 100% (blue) excess of RbBr are provided in Figure 2A. The excitonic peaks corresponding to the quasi-2D domains with $n = 1$, $n = 2$, and $n = 3$ are clearly seen in the absorption spectra around 405, 425, and 450 nm, respectively, and are consistent with the positions previously reported [26]. This demonstrates that the perovskite film does not consist of a single domain but contains a variety of domains with different $n$ values, which agrees with the previous publications [44,45]. In contrast with the absorption spectra, a single band strongly dominates the PL spectra of all three samples, whereas the additional bands observed at shorter wavelengths are much weaker. This means that, although the layers contain multiple $n$ domains, the emission takes place predominantly from a single domain with a certain $n$, which means that the exciton transport to this particular domain is fast and efficient. The peak wavelength of the PL band (470–490 nm) indicates that the emission originates from $n = 4$ or thicker domains [26,46]. The absorbance and PL bands of the domains become closely spaced with the increasing $n$ number, which makes the assignment of the particular $n$ value difficult.

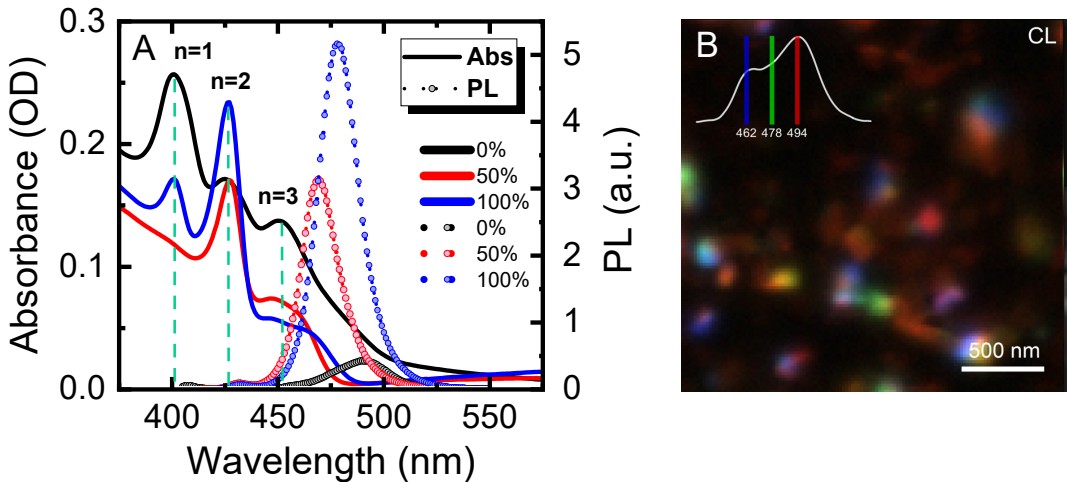

**Figure 2.** (**A**) Absorbance (solid lines) and PL (dashed lines) spectra of quasi-2D Rb–Cs-alloyed lead bromide perovskites with varying excesses of RbBr. Excitonic peaks in the absorption spectra are labelled, indicating the corresponding quasi-2D domains. (**B**) Cathodoluminescence image of the sample with 0% excess RbBr. The image is artificially colored in RGB primaries and displayed in its original 128 × 128-pixel grid. The correspondence between the luminescence wavelength and the pixel color is shown in the inset. The scalebar is shown on the bottom right.

The use of an excess of RbBr leads to a significant increase in the $n = 2$ phase at the expense of the $n = 1$ phase, as can be clearly concluded from the absorption spectra presented in Figure 2A. In addition, the PL intensity increases, the band shifts to shorter wavelengths, and its width becomes smaller with the increasing amount of RbBr, confirming the results reported previously [26]. The PL band parameters are listed in Table 1. The broader PL band of perovskite without an excess of RbBr might indicate that the emission occurs not from the domains with a single *n*, but rather from multiple domains.

**Table 1.** Wavelength and photon energies of peak positions and full width at half-maximum (FWHM) of PL bands in samples with different amounts of excess RbBr content.

| Excess of RbBr Content | Peak PL Wavelength (nm) | Peak PL Photon Energy (eV) | Spectral Width at FWHM (nm) |
|---|---|---|---|
| 0% | 490 | 2.53 | 30 |
| 50% | 469 | 2.64 | 21 |
| 100% | 478 | 2.59 | 22 |

Figure 2B shows a cathodoluminescence image of the 0% sample. The images for the 50% and 100% excess RbBr samples are shown in Figure S1, Electronic Supplementary Information (ESI). The cathodoluminescence spectra were measured in every pixel of the image, and then the pixels were colored according to peak luminescence wavelength. The image shows that the sample is heterogeneous and is composed of particles emitting at different wavelengths. Moreover, the density of the emitting particles is low, which is a likely consequence of efficient exciton funneling into the emitting domains that make up only a minor fraction of the perovskite film.

*3.2. Transient Absorption Spectroscopy*

To investigate the dynamics taking place in the perovskite layer after excitation, we performed TA spectroscopy measurements. The capability of probing the density of nonemitting, nonequilibrium carriers with a time resolution of <1 ps is a valuable advantage of TA spectroscopy. An excitation wavelength of 415 nm was used, which means that only $n = 2$ and thicker domains were excited. The TA spectra of the perovskite films containing

0%, 50%, and 100% of excess RbBr are shown in Figure 3. The spectra of the 0% film reveal 3 negative peaks at 455, 475, and 493 nm. In general, negative peaks can be a result of either ground-state bleaching (GSB) or stimulated emission (SE), both of which are caused by the exciton that interacts with the probe pulse. The shortest-wavelength (455 nm) band is the result of the GSB of the $n = 3$ domain, as it coincides with the absorption band of this domain (see Figure 2A). The 475 nm band is assigned to the GSB of the $n = 4$ domain since the wavelength approximately corresponds to the band gap of this domain [28]. The 493 nm band consists of overlapping GSB bands and the band of SE. The latter is expected since the observed luminescence band is also present at 493 nm (see Figure 2A). As the separation of excitonic absorbance and PL bands becomes progressively smaller with increasing $n$-values, the assignment of the 493 nm band is complicated. Therefore, we assigned the signal at 493 nm to the $n \geq 5$ domain. In addition to the negative signals, there is a positive TA band at 440 nm due to the excited-state absorption (ESA). In contrast with the 0% sample, the 50% and 100% samples show GSB bands at 430 nm, corresponding to the $n = 2$ domain, which agrees well with the much more intense $n = 2$ absorption bands for the 50% and 100% samples in Figure 2A. In addition, the higher wavelength bands in the 50% and 100% samples are shifted to shorter wavelengths in comparison with the 0% sample, namely to 468 nm (50% sample) and 473 nm (100% sample), which agrees well with the shift of the PL peaks in Figure 2A. The bands likely originate from combined GSB and SE signals from the $n = 4$ domain. The slight difference in wavelength could be explained by a higher ratio of $Rb^+/Cs^+$ in the particular domain that gives the band at 468 nm, even though the average $Rb^+/Cs^+$ ratio is higher in the 100% sample. It is known that a higher content of $Rb^+$ increases the band gap of quasi-2D perovskites with the same $n$ [26]. Furthermore, a higher excess of RbBr might limit the formation of $n > 4$ domains, which leads to the shift of the GSB band to 468-473 nm from its position at 493 nm in the 0% sample.

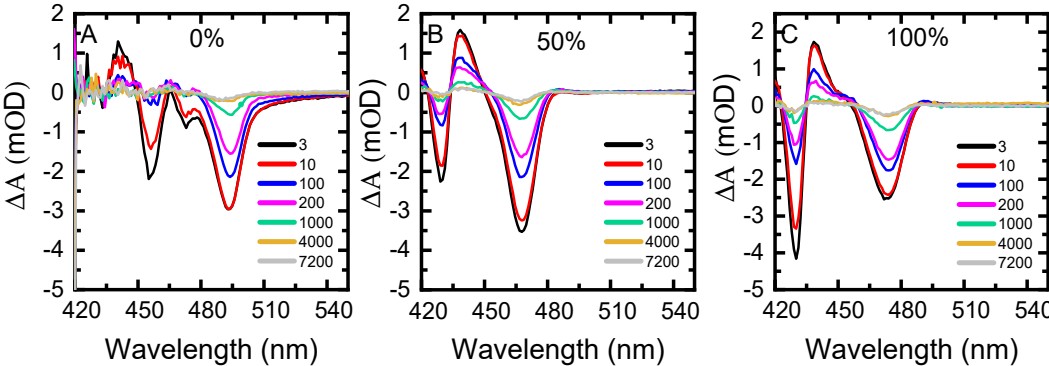

**Figure 3.** Transient absorption spectra of perovskite samples with no excess (**A**), a 50% excess (**B**), and a 100% excess (**C**) of RbBr at different delay times between pump and probe pulses. The delays are shown in the legend.

The TA experiments for the sample with a 100% RbBr excess were repeated at 2 lower temperatures, 50 °K and 200 °K (see Figure 4). If the observed TA features were caused by free carriers, lowering the temperature should have led to significant changes in the TA spectra and kinetics [29]. Meanwhile, the transient absorption spectra at 50 °K and 200 °K (Figure 4A,B) show only minor differences compared to the spectra at room temperature (297 °K, Figure 4C). The TA kinetics at different temperatures (Figure 4D) are nearly identical. This is an important indication that the system of nonequilibrium carriers in quasi-2D perovskites studied in the current work is dominated by excitons.

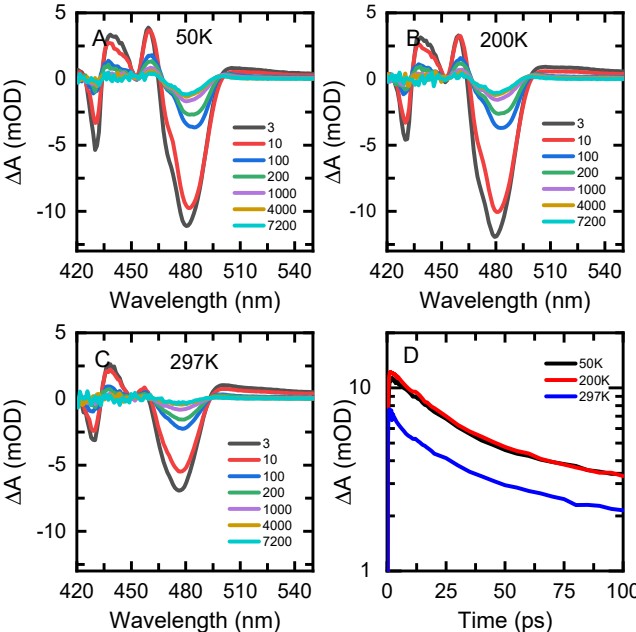

**Figure 4.** Transient absorption spectra of perovskite sample with a 100% excess of RbBr at 50 °K (**A**), 200 °K (**B**), and 297 °K (**C**) with different delays between pump and probe pulses (indicated) and the time evolution of the transient absorption signal at 475–480 nm at different temperatures (indicated) (**D**).

Next, we investigated the exciton-density dependence of the processes observed in quasi-2D perovskites. For this purpose, we performed TA experiments at different excitation-pulse energy densities from 1.43 to 28.6 µJ/cm$^2$. Figure 5 shows the time evolution of the TA signal at different excitation-energy densities in samples with 0%, 50%, and 100% excess RbBr in 2 main GSB bands. First, it is evident that the signal decrease is faster at shorter wavelengths corresponding to the GSB band of either the $n$ = 3 domain (0% RbBr excess sample) or the $n$ = 2 domain (50% and 100% samples). Second, for the majority of samples, the decrease of the TA signal within the first 100 ps is insignificant at the lowest excitation-pulse energy density of 1.4 µJ/cm$^2$, whereas a fast component appears at higher excitation intensities. The change in the TA time evolution at increasing excitation intensities indicates that the second-order processes are taking place. Since the carrier dynamics are dominated by excitons, as confirmed by the temperature-dependent data in Figure 4, the most likely second-order process behind the TA features presented in Figure 5 is exciton–exciton annihilation. Furthermore, several TA kinetics show an increase of the TA signal, as shown in Figure 6. This is a clear signature of the energy transfer that occurs from thinner quasi-2D domains to the emitting $n \geq 4$ domain. It is worth noting that energy transfer between different domains within 1 ps has been previously reported [26], whereas we observed that the energy transfer also occurs on a much longer time scale (within 50 ps). The rising part of the signal due to energy transfer is observable only at intermediate excitation-pulse energy densities, while the exciton–exciton annihilation dominates at high densities. Finally, increasing the RbBr excess in the perovskite layer does not lead to significant changes in the time evolution of the TA spectra.

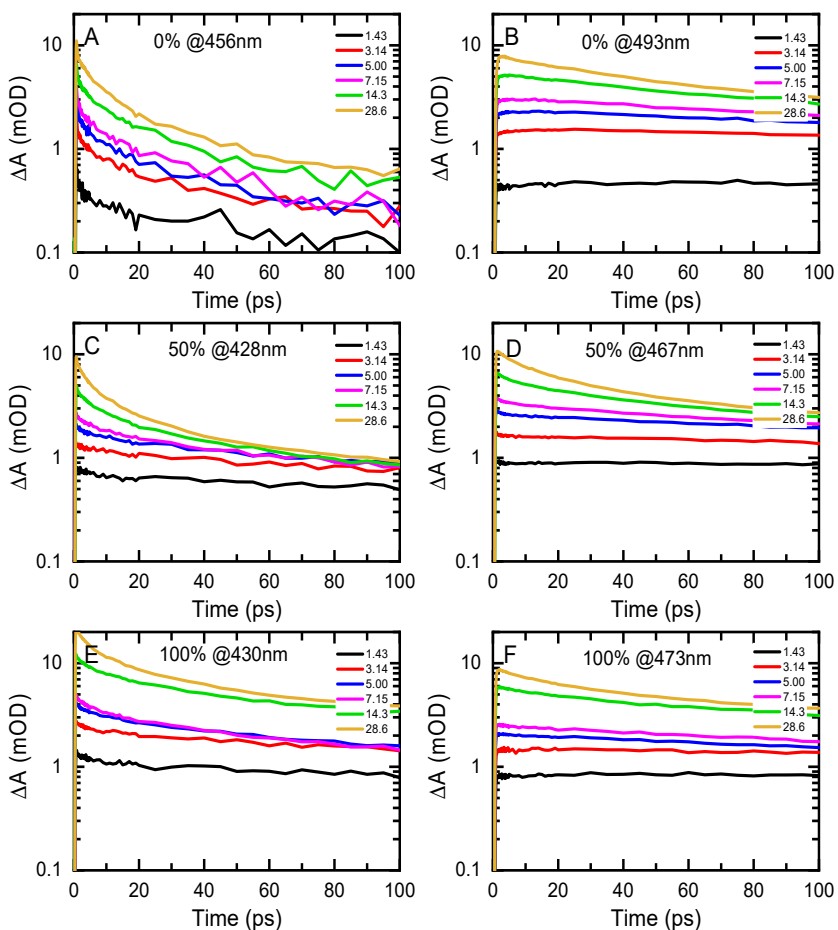

**Figure 5.** Time evolution of transient absorption in perovskite films with a 0% (**A**,**B**), 50% (**C**,**D**), and 100% (**E**,**F**) excess of RbBr at varying excitation-pulse energy densities shown in µJ/cm². The selected wavelengths correspond to the peak of the GSB band from the $n$ = 3 domain in the 0% sample (**A**), $n$ = 2 domain in the 50% and 100% samples (**C**,**E**), and the emitting $n \geq 4$ domain (**B**,**D**,**F**).

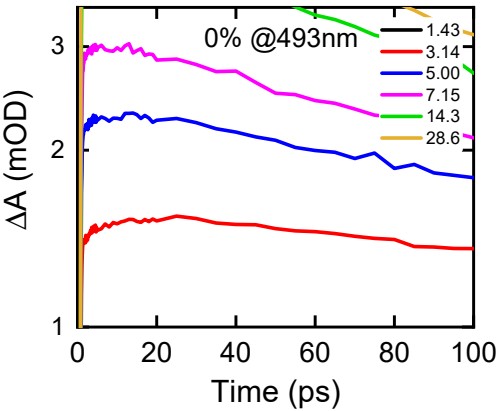

**Figure 6.** Time evolution of transient absorption at the peak of GSB band of the perovskite film without excess RbBr at intermediate excitation-pulse energy densities.

Finally, we investigated the exciton dynamics in Rb–Cs quasi-2D perovskites in more detail. The density of excitons ($n$) is described by the following kinetic equation:

$$dn/dt = -\gamma n^2 - kn - \eta n, \tag{1}$$

where $t$ is the time, $\gamma$ is the rate constant of exciton–exciton annihilation, $k$ is the rate constant of exciton decay, and $\eta$ is the rate constant for the energy transfer between quasi-2D domains. For high excitation intensities, the first term $\gamma n^2$ is dominant, and the remaining terms can be neglected. In this case, the solution of the simplified kinetic equation is as follows:

$$1/n - 1/n_0 = \gamma t, \tag{2}$$

where $n_0$ is the exciton density immediately after excitation. Therefore, a linear dependence of $1/n$–$1/n_0$ on time is expected, provided that the exciton–exciton annihilation is the predominant decay path. Calculating the exciton density in a heterogeneous sample of this type is not possible. Therefore, we calculated the relative absorption bleaching $N$, which is proportional to the exciton density. $N$ was calculated as $N = \Delta A/A$, where $\Delta A$ is the TA signal, and $A$ is the absorbance of the perovskite film at the particular wavelength, at which the signal was recorded. The induced absorbance $\Delta A$ measured in the TA experiments was proportional to the number of excitons probed in the sample. However, the exciton dynamics depend not on the number of excitons, but on their density. To estimate it, the domain volume must be known. The volume can be inferred from the steady-state optical absorption, which is proportional to the domain volume. Therefore, the relative bleaching $N = \Delta A/A$ is proportional to the exciton density in the domain of interest. Finally, $1/N$–$1/N_0$ was plotted as a function of time for different excitation-pulse energy densities. The curves were similar, so we averaged them to obtain the best signal-to-noise ratio.

The result is shown in Figure 7, where the dependences for the $n = 2$ GSB bands (Figure 7A, 50% and 100% samples) and $n \geq 4$ GSB bands (Figure 7B, all 3 samples) are presented. In the case of the $n = 2$ bands, the plotted dependence is affected by the energy transfer to thicker quasi-2D domains at early delays, but it becomes linear afterwards. The linear dependence shows that exciton–exciton annihilation dominates the signal. In contrast, the $n \geq 4$ GSB band shows a linear dependence throughout the entire time range. It is worth noting that the absolute values of $1/N$–$1/N_0$ are substantially larger for the $n = 2$ band. This is an indication that exciton–exciton annihilation in the thin $n = 2$ domain is much faster than that in the thicker $n \geq 4$ domain. The faster annihilation could be the consequence of faster exciton diffusion in thinner domains [47]. This phenomenon likely facilitates the rapid exciton transfer from low-$n$ to thicker emitting $n \geq 4$ domains. The observed faster exciton annihilation in the $n = 2$ domains is consistent with previous observations obtained using TA spectroscopy and microscopy which showed a decrease in the exciton–exciton annihilation rate with the increase in the $n$-value of the quasi-2D domain [34]. The result leads to the conclusion that higher exciton densities might be achieved in thicker, higher-$n$ domains, which is important for PeLEDs with maximum brightness.

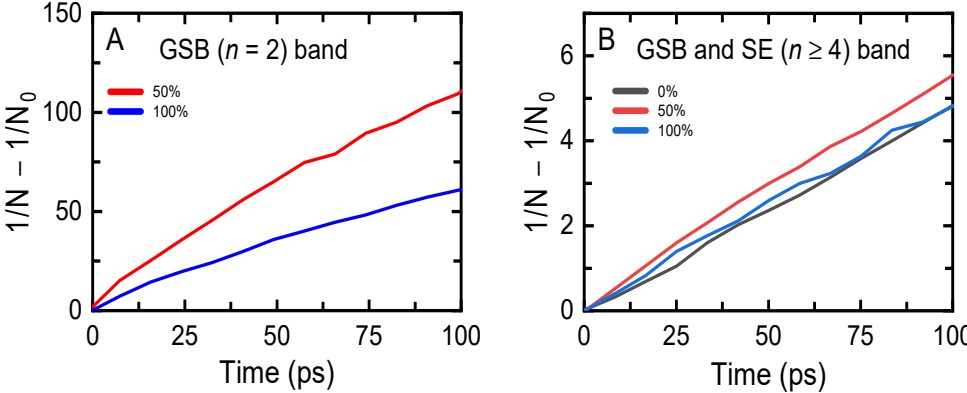

**Figure 7.** Time evolution of the inverse relative bleaching $(1/N)$ obtained from the TA signal in the GSB bands from the $n = 2$ domain in samples with a 50% and 100% excess of RbBr (**A**) and emitting $n \geq 4$ domains in all samples (**B**). The detection wavelengths were approximately 430 nm (**A**) and 470–490 nm (**B**).

## 4. Conclusions

In summary, we have investigated the exciton dynamics in Rb–Cs-alloyed quasi-2D perovskite using TA spectroscopy. We have studied 3 samples with excess RbBr contents between 0% and 100%. According to the absorption spectra, the structure of the samples is dominated by $n = 1$ and $n = 2$ domains. Our results support the hypothesis that the system of nonequilibrium carriers in such perovskites consists predominantly of excitons. We have observed the transfer of the excitons from the $n = 2$ domains to emitting $n \geq 4$ quasi-2D domains, which have a larger thickness. Additionally, we have seen evidence of an excess of RbBr preventing the formation of $n = 1$ domains. Moreover, we have shown that exciton–exciton annihilation is the dominant exciton decay pathway at higher elevated excitation-energy densities. This process in thinner $n = 2$ domains is substantially faster than that in $n \geq 4$ domains. This result suggests that quasi-2D perovskites with higher $n$-values are favorable for perovskite lasers or bright perovskite LEDs, which require high exciton densities.

**Supplementary Materials:** The following supporting information can be downloaded at: https://www.mdpi.com/article/10.3390/photonics9080578/s1, Figure S1: Cathodoluminesence images of perovskite films with 50% and 100% excess of RbBr.

**Author Contributions:** Conceptualization, A.V. and V.G.; investigation, L.A., Ž.P. and V.J.; writing—original draft preparation, L.A. and A.V.; writing—review and editing, A.V. and V.G.; supervision, V.G., R.A., G.T. and A.V.; funding acquisition, A.V. All authors have read and agreed to the published version of the manuscript.

**Funding:** This research was funded by the European Social Fund under the No 09.3.3-LMT-K-712 "Development of Competences of Scientists, other Researchers and Students through Practical Research Activities" measure.

**Institutional Review Board Statement:** Not applicable.

**Informed Consent Statement:** Not applicable.

**Data Availability Statement:** The data is available upon reasonable request. The data presented in this study are available on request from the corresponding author.

**Conflicts of Interest:** The authors declare no conflict of interest.

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
