# Peer review of "Exciton Transfer Dynamics and Annihilation in Rubidium–Cesium-Alloyed, Quasi-Two-Dimensional Perovskite"

_photonics, doi:10.3390/photonics9080578_

Round 1

Reviewer 1 Report

In this manuscript, the authors investigated the exciton dynamics in Rb-Cs alloyed quasi-2D perovskites using transient absorption spectroscopy at different temperatures and found that the structure of the samples is dominated by (n = 1 and n = 2) domains. Moreover, they further show that exciton-exciton annihilation is the dominant exciton decay pathway at higher elevated excitation energy densities. This process is thinner (n = 2) and substantially faster than in (n ≥ 4) domains. I think the manuscript and corresponding data after revision could meet the requirement of Photonics, but I still have suggestions that might further improve the quality.

1. It would be better if the authors could provide characterization data of PEA2(Rb0.6Cs0.4)2Pb3Br10 perovskite materials with different excess of RbBr, such as SEM, AFM, PLQYs, and PL lifetime. This will be a good part of the draft.

2. The schematic quality shown in Figure 1 is simple and unclear. The authors should improve them before the publication of this work.

3. In Figure 3c, an enhanced negative peak at 470 nm can be observed. The origin of these enhanced negative peaks is suggested to be clearly explained.

4. For perovskite material, photostability and thermal stability have always been the focus of attention, so it is suggested to increase the work on strength (Nanoscale, 2020, 12, 6403).

5. There are many awkward usages and errors in the manuscript. For example, many of the peak values claimed in the draft are inconsistent with those reflected in the Figure. Therefore, I suggest the authors further improve quality, including writing and logic.

6. The crystal structure characterization is essential for perovskite material. Please provide the corresponding XRD characterization.

7. The n value of two-dimensional perovskite films can be adjusted by adding different amounts of RbBr. I’d like to see more explanations and further evidence.

8. For the cathodoluminescence image, 50%, and 100% samples are missing, and please add the corresponding characterization.

Reviewer 2 Report

In this manuscript, the authors exploited transient absorption spectroscopy to investigate exciton dynamics in quasi-2D perovskites with and without an excess of RbBr. They demonstrated that exciton transferred from thinner (n = 2) domains to thicker (n >= 4) domains. In addition, they showed that exciton-exciton annihilation in thin domains is much faster than that in thick domains. However, minor revisions are needed.

Suggestions to authors for improving the manuscript:

1. “migration” in the title is misleading. There are not many details about exciton diffusion, disassociation, or recombination at different sites. The manuscript is focused on the transfer dynamics of the exciton.

2. What’s the pulse duration of pump pulses? What’s the pulse duration and wavelength of probe pulses?

3. In line 215, the authors claimed “higher content of Rb+ increases the bandgap”. 100% sample has a higher concentration of Rb+, indicating it should have a shorter PL wavelength compared to the 50% sample. Why is the observation opposite?

4. Band positions in Figure 3 agree with the PL peak position in Figure 2A at 490 nm for 0% sample, 470 nm for 50% sample, and 480 nm for 100% sample, respectively. Why do their intensities/depths in Figure 3 not have a similar trend shown in Figure 2A, i.e. I(100%) > I(50%) > I(0%)?

5. It would be better to plot the first 50 ps with three moderate intensities in Figure 6 to clearly show the energy transfer time scale.
